# Predicting severe stunting and its determinants among under-five in Eastern African Countries: A machine learning algorithms

Halid Worku Jemil◉[1]*, Sonia Worku Semayneh[2], Altaseb Beyene Kassaw[3], Kassahun Dessie Gashu[4]

1 Department of Health Informatics, College of Medicine and Health Science, Wollo University, Dessie, Ethiopia, 2 Department of Oncology, Addis Ababa University College of Health Science, Tikur Anbessa Specialized Hospital, Addis Ababa University, Addis Ababa, Ethiopia, 3 Department of Biomedical Science, College of Medicine and Health Science, Wollo University, Dessie, Ethiopia, 4 Department of Health Informatics, College of Medicine and Health Science, Gondar University, Gondar, Ethiopia

* halidworku288@gmail.com

## Abstract

### Introduction

Severe stunting is one of the primary public health challenges in LMIC including Eastern African Countries, which affects millions of children. In addition, it was a major contributor for mortality and related complication of children aged under five. However, there is limited study conducted severe form of stunting by employing Machine learning (ML) in Eastern African Countries. Therefore, our study was demonstrated to predict and identify its major determinants using ML algorithms, furthermore, to improve model explainablity. Our study used Shapley Additive explanations (SHAP) and ARM to identify the determinants of severe stunting among under-five.

### Methods

cross-sectional study was conducted using DHS data from 2012–2022 in East Africa. 136,074 children were the source populations, and 76,019 children were the study population. Data were analyzed using Python version 3.7 and R version 4.3.3 for data preprocessing, modeling, and statistical analysis. Model performance was evaluated using accuracy and AUC. Furthermore, the SHAP analysis and ARM was used to further explain and interpret the determinants of severe stunting among children under five.

### Results

The Random Forest performed the best in this analysis, with an accuracy of 87% and an AUC score of 0.83. The analysis indicated that women's who do not practicing exclusive breastfeeding (SHAP value = +0.41), being from Burundi (SHAP

**Data availability statement:** All relevant data are within the manuscript and its Supporting Information files.

**Funding:** The author(s) received no specific funding for this work.

**Competing interests:** The authors declare that they have no conflict of interest exist.

**Abbreviations:** AUC: Area under the Curve; BMI: Body Mass Index; CI: Confidence Interval; DHS: Demographic and Health Survey; HAZ: Height for Age Z-score; KNN: K-nearest neighbor; ML: Machine Learning; RF: Random Forest; ROC: Receiving Operating Characteristics; SHAP: Shapely Additive Explanation; SSA: Sub-Saharan Africa; SVM: Support Vector Machine; SMOTE: Synthetic Minority Oversampling Techniques; UNICEF: United Nation International Children's Emergency Fund; WHO: World Health Organization; XGB: Extreme Gradient Boosting.

value = +0.04), children being underweight (SHAP value = +0.25), lived in poor household (SHAP value = +0.40), child gender being male(SHAP value = +0.23), mothers height being short (SHAP value = +0.03), mothers being underweight (SHAP value = +0.18), child size at birth being small (SHAP value = +0.21), women's being delivered in home(SHAP value = +0.07), mothers education being primary (SHAP value = +0.20), unimproved toilet (SHAP value = +0.06), distance to health facility being a big problem (SHAP value = +0.02), were associated with increase the risk of severe stunting among under five.

## Conclusion

The Random Forest was the best-performing model for predicting severe stunting in Eastern African countries. To decrease the effects of severe stunting, integrated interventions should provide support for mothers with lower socioeconomic conditions, strengthen maternal education, empower women to practice exclusive breastfeeding, encourage facility deliveries, increase access for households to sanitary facilities, provide education on personal and environmental hygiene, provide mothers with information on the importance of complementary feeding for children as well as for the mothers, and provide near health facilities for mothers and essential care services.

## Introduction

Severe stunting is a critical form of malnutrition, causing extreme growth impairment and development in children. It is mainly caused by a chronic lack of adequate dietary intake, repeated infection, and inadequate psychological stimulation, resulting in several health and developmental problems [1–3]. It results in destructive health consequences, including impaired defense against infection, growth retardation, delayed recovery from infection, poor intelligence quotient, high anxiety and depression, and increased susceptibility to disease. It also causes poor economic productivity and poor educational attainment [4–7].

According to the 2022 WHO/UNICEF/World Bank Group report, globally 148.1 million under five and 37 million children were stunted and severely stunted respectively [8]. Stunting is most prevalent in developing and low-income countries, the highest prevalence were found in Southern Asia at 20% and SSA at 30% [5,9]. In SSA, 10.5% of children were severely stunted [10]. The prevalence is higher in Eastern African Countries compared to other SSA countries, with Burundi (24.8%), Ethiopia (17%), Tanzania (11.8%), Zambia (11.7%), Mali (10.5%), Rwanda (9%), and Zimbabwe (8%) [11–13].

Based on previous studies, several factors are assumed to contribute to stunting among under five, including socio demographic/household related factors such as place of residence, breastfeeding practice, sanitation and hygiene, distance to health facilities, wealth index, educational status, media exposure, and maternity/child

related factors include maternal weight, ANC and PNC visit birth type, child sex, child size at birth, birth order, and mother's height [13–19].

Different interventions and strategies have been conducted to alleviate the burden and consequences of severe stunting. The WHO adopted a plan to reduce all forms of malnutrition in Africa by strengthening laws and food safety rules, employing financial initiatives to encourage healthier food options, and incorporating vital nutrition initiatives into healthcare delivery systems [20]. Similarly, the Africa regional nutritional strategy 2015–2020 also emphasizes overcoming severe stunting [21–23]. In addition, Sustainable Development Goals (SDGs) were conducted as goal 2.2 stated that zero hunger, which is implemented to get rid of malnutrition such as stunting and wasting among children under five years old, will be achieved by effectively providing the essential nutrition for adolescent girls, pregnant and lactating mothers, and older persons by 2030. However, according to the SDG progress reports, current efforts are insufficient to meet the global target of reducing the prevalence of malnutrition by 40% by 2025. Recent data from various reports even indicate an increase in severe stunting [7,13,24,25].

Different studies was conducted on stunting and its associated factors However, there is limited research conducted on the severe form of stunting. In addition, most of them were from health record data and hospital settings and country level, which makes it impossible to generalize findings [13,19, 26–43]. In addition, almost all studies use traditional regression analysis to identify the risk factors, which are ideally the risk of overfitting with many predictors, inability to handle large datasets, inability to handle complex interactions like non-linear computation, and struggle with larger data [44,45]. In contrast to this, ML algorithm can enhance the accuracy of prediction than traditional regression models [46]. ML has the ability to handle multiple datasets, complex interactions like non-linear computation, and healthcare organizations can build accurate models that can help to accurately predict and estimate to enhance patient health outcomes [47,48]. Therefore, this study utilized ARM and SHAP analysis to identify the predictors of severe stunting from the best-performing model by using DHS data collected from 2012 to 2022 in Eastern African countries among children under five years old.

## Methods

### Data source and study setting

This study employed cross-sectional design, drawing data from the DHS collected from 2012 and 2022. This study utilized 12 Eastern African countries presented on the map (S1 Fig 1 in S1 File) namely Burundi (2017), Ethiopia (2016), Rwanda (2019), Uganda (2016), Comoros (2012), Zambia (2018), Tanzania (2022), Mozambique (2022), Madagascar (2021), Zimbabwe (2015), Kenya (2022), and Malawi (2016) (S1 Table 1 in S1 File).The data is available in the Measure DHS website https://www.dhsprogram.com/data/.

### Population

**Source population.** All children who lived in 12 Eastern African countries were the source population.

**Study population.** All children who lived in 12 Eastern African countries and whose mothers/caregivers were present in the household during the enumeration period were the study population.

**Inclusion and exclusion criteria.** Children under the age of five whose mother/caregivers were present in the household during the enumeration period were included in the study. Whereas, children under the age of five years with missing or incomplete records or flagged cases (outliers) were excluded from the study [49].

**Sampling method and sample size determination.** DHS uses a standardized and validated questionnaire. It used a two-stage stratified sampling technique to select representative study participants. To begin with, the Enumeration Areas (EAs) were chosen using a probability method that was aligned with the size of each area, making sure the selection was done independently in every sampling group. In the next phase, homes were selected in a systematic way. The main DHS indicators were collected in each DHS [50]. All the detailed information for the survey (such as the sampling method,

the determination of the sample size, and the data collection procedure) is available in Demographic and Health Survey reports from the Measure DHS program website https://www.dhsprogram.com.

Weighted 136,074 children under five were included in this study. From them 59,505 children whose caregivers/mothers were not present during the enumeration period, and 550 children with implausible or flagged cases were removed. Finally, this study used a weighted sample of 76,019 children under five years for the analysis, namely Burundi (6048), Ethiopia (8855), Rwanda (3809), Uganda (4423), Comoros (2387), Zambia (8746), Tanzania (4807), Mozambique (3733), Madagascar (5778), Zimbabwe (4957), Kenya (17327), and Malawi (5149). The detailed flowchart for the selection of study participants presented in (S1 Fig 2, and S1 Table 1 in S1 File).

### Study variables

**Outcome variable.** The outcome variable for this study was severe stunting.

**Independent variables. Maternal/household related variables**: Mother's education, Marital status, Mother's occupation, maternal-Height, maternal-BMI, household media exposure, Father's education, types of toilets, health insurance coverage, household wealth status, Source of Fuel, Family size, Sex of Household head, Time to get drinking water, Country, Distance to health facility, Place of residence.

**Child-related/maternity-related variables:** Sex of child, Age of child, Birth order, and Child size at birth, Underweight status of the child, place of Delivery, Modes of Delivery, Age at first birth, Types of birth, ANC visits during pregnancy, PNC care, exclusive breastfeeding for six months.

**Operational definition. Severe stunting** was dichotomized into a binary category based on height-for-age Z-score. Children were considered severely stunted if their Z-score $< -3$ SD and not severely stunted if their Z-score $>= -3$ SD based on the WHO child growth reference standard [12].

**Maternal BMI** was categorized into three groups based on her weight relative to her height: if BMI $< 18.5$, labeled underweight; if BMI is between 18.5 and 24.9, labeled as normal; if BMI $> 25$, labeled as overweight/obese [51].

### Data quality

The DHS program employs a data file that appropriately represents the population investigated by utilizing a policy of editing and imputation to resolve such difficulties [50]. Similarly, it uses a different procedures to improve data quality. After all questionnaires had been entered, each survey was double-entered, and the two datasets were compared to ensure accuracy. Discrepancies and inconsistencies in the data were thus rectified, and certain missing values were also filled in. The particulars of quality assurance are available at https://www.dhsprogram.com/data/ [52].

### Data management and analysis

Before doing the analysis, weight adjustments were applied to handle the complexity of sampling design and to ensure representativeness using Stata version 17 for. We adjusted the data for both the outcome and predictors using DHS sample weights (v005/1,000,000) in all descriptive analyses. This helps to correct for unequal probability of selection and to ensure a national representativeness of samples. Additionally, the data were preprocessed, missing values were managed before conducting analysis. This study used a ML approach based on Yufeng Guo's 7 steps of ML and from the frameworks of a previous study. The seven steps employed in the data management and analysis include Data collection, Data Preprocessing, Model selection, Model training, Model evaluation, Hyperparameter tuning, and Making prediction, and for this study, ML algorithms were implemented using Python version 3.7 on Jupiter notebook [53,54].

### Data collection and preprocessing

For this study, kids' data (KR) from the DHS conducted from 2012 to 2022 were utilized, which are available in the Measure DHS program website https://www.dhsprogram.com/data/. The first process of making data suitable for analysis

includes weight adjustments were applied to account for the complex sampling design, ensuring representativeness, data cleaning, feature engineering, Dimensionality reduction, and data splitting.

### Data cleaning

Data was extracted, recoded, and operationalized based on DHS and WHO guidelines by using STATA version 17, and multicollinearity among the independent variables was checked by using VIF (Variance Inflation Factor) values. The VIF values for each independent variable were < 2, which confirms that there was no multicollinearity between variables. Missing record was impute by KNN which is effective for both categorical and continuous variables, it is flexible and suitable to handle large and complex datasets (>50,000) [55,56]. Additionally, the KNN Imputation showing the best performance (highest Accuracy: 0.897088 and F1-Score: 0.880405), followed by MICE and Mean/Mode imputation (S1 Table 2 in S1 File).

### Feature engineering and dimensionality reduction

It is the process of generating new variables to enhance ML efficacy and improve model performance [57]. Two methods of feature engineering were used, one-hot encoding for nominal and label encoding for ordinal feature.

Dimensionality reduction (DR) is the process of reducing the attributes used to build the model for prediction. Fewer variables can lead to simpler, more efficient models with improved performance on unseen data [58]. Feature selection involves prioritize the best relevant variables for predicting the dependent variable based on their statistical relationships. To detect the best predictors of severe stunting, we used the Boruta feature importance technique [54]. Likewise, large datasets with many attributes often not important for model development, decrease accuracy and performance [59]. Non-important variables were rejected by the algorithm; and dimensionality reduction involves decreasing the number of input features to increased model efficiency, improve model performance.

### Data splitting and model selection

Data was split into training and testing sets by allocating 80% (54,111) of the data for training and 20% (15,204) of the data for testing to evaluate the model. This study employed a 10-fold CV technique to train and test data since 10-fold CV prevents data waste, and enables model to train on most characteristics and a smaller amount of data for testing. It is an efficient way to enhance model performance [60].

After the data was prepared and divided into training and testing sets, appropriate models were selected to carry out the training by considering the nature of the dependent variable and the task to be performed. Hence, the dependent variable was binary categorical, and the task was classification. The right models were chosen, such as RF (RF), Logistic Regression (LR), Decision Tree (DT), extreme gradient boosting (XGB), K-nearest neighbor (KNN), AdaBoost, support vector machine (SVM), and CatBoost performed on Python version 3.7 using Jupyter Notebook [19,26,28,58,61].

### Balancing the data

To fix the class imbalance in the dataset and make sure the minority class is well represented, synthetic samples were made to improve the minority class without overfitting or letting the majority class take over the learning process for the training sets [62,63]. Thus SMOTE was one of the several data balancing methods that are available. Hence, SMOTE was adaptable to large datasets, flexible, and robust than other methods [64].

### Model building/training and performance evaluation

After choosing the right model, the models were trained on both balanced and unbalanced data; the best predictive model was chosen after comparing the models on balanced training data. The model's performance was measured by using performance metrics such as sensitivity, specificity, accuracy (Table 1), and AUC, which were employed to illustrate how well

the models perform in terms of predicting severe stunting [65]. Our study primarily used the AUC score to compare and select the best models, since AUC is the best method, particularly for binary classification techniques [66].

As illustrated in Table 1 two by two tables, the following formula were used to calculate.

1. Accuracy = TP + TN/TP + TN + FP + FN

2. Sensitivity = TP/TP + FN

3. Specificity = TN/TN + FP

4. Precision = TP/ TP + FP

Utilizing the above metrics, the study comprehensively evaluated the performance of each predictive model in terms of overall correctness, accurate positive predictions, and identification of positive instances, balanced measure, and discriminatory ability.

### Hyperparameter

A hyperparameter is an external manipulation to the model whose value must be set by the user [58]. The selected model was optimized with the best parameters by applying the Randomized search technique with 10-fold cross-validation on the specified search space with one hundred trials. Since these techniques are more efficient when we deal with complex models and larger datasets than other techniques, by using Python version 3.7 on the Jupiter notebook [67].

### Making predictions and model interpretability using SHAP

The last step of the ML process involves figure out the dependent variable by using the independent variables. Whether a child was severely stunted or not severely stunted was identified by using the best and optimized mode using Python v3.7 [68]. SHAP was an advanced model interpretation techniques which is used explain and interpret ML model. Additionally, SHAP improves the interpretability and clarifies complex models [68]. The SHAP explainer was build based on an optimized and best-achieved model to further explore the global and local explanations for the test set on Python v3.7. SHAP global explanation was the first step to explain the model. It explains the overall behavior of the model in global fashion, it includes such as summary plot and a Beeswarm plot. The summary plot visualize the mean absolute SHAP value which was considered to show impact of individual predictions on overall models output. In addition, it which shows the importance of each feature's influence on model prediction in a global fashion. Beeswarm plot showing the average impact of each characteristic and direction of the prediction, either positive or negative [68]. On the other hand, local explanations focus on interpreting individual predictions made by an ML model. SHAP provides such local interpretability by attributing each feature's contribution to a specific prediction. The model generates distinct predictions for each instance, and SHAP analysis was leverage to explain the predictions locally by dividing down how each feature contribute to the prediction. Visualization tools like waterfall plot were most commonly used to explains these local explanations, by giving directions to how each features influence either positively or negatively to the prediction based on a baseline value [69].

**Table 1. Evaluation metrics for the prediction of severe stunting in Eastern African countries, 2012–2022.**

| Classes | Predicted class | | |
|---------|------|------|------|
| | **Yes** | **No** | **Total** |
| Yes | TN | FP | TN + FN |
| No | FN | TP | FN + TP |
| Total | TN + FN | FP + TP | TN + FN + FP + TP |

## Association rule mining

Association rule mining (ARM) was conducted based on the apriori algorithm, executed with a rules package of the R software (version 4.3.3), for the identification of specific categories of predictor variables associated with severe stunting. Association rules expressed in the form of IF–THEN rules, facilitate the detection of strong features during rule induction. Unlike statistical significance testing, ARM is interested in finding strong and frequent associations between variables by taking into account measures of interestingness that are highly correlated with the effect size of observed patterns [70].

An association rule is described as A→B, where A (antecedent) appears on the left side of the rule and B (consequent) is on the right side. The rule indicates that the presence of A leads to being associated with the presence of B. ARM searches extensively through datasets in order to discover frequent patterns, relationships, and value associations among variables, gaining valuable insights into the complex interdependencies between the data. The result of association rule mining was presented with their lift values, a lift measures the degree to which the consequent is more likely to happen, and it provides evidence for the relationship above chance [71]. A lift of 1 indicates that A and B are independent (no relationship). A lift value greater than 1 indicates a positive relationship; the occurrence of A will make it more likely for B to occur. A lift value less than 1 indicates a negative relationship; the occurrence of A will make it less likely for B to occur [72]. The data preprocessing and analysis workflow is summarized in (S1 Fig 3 in S1 File).

## Ethical approval and consent from participants

Our study was used a secondary data from DHS after we granted permission through proper registration to access the data from the DHS website https://www.dhsprogram.com/data/. The data for this study was downloaded after authorization, approval and consent from the DHS committee.

## Results

### Background characteristics of participants

**Child-related/maternity-related characteristics.** A total of 76,019 children under five years were included in the study; of them, two-thirds, 45,855 (60.32%), were aged between 24 and 36 years. More than half of the children, 42475 (55.87%), had an average size at birth. About 57057 (75.06%) of participants were delivered to health facilities. More than half of the participants, 57660 (75.85%), experienced their first birth at a younger age (9–21 years). About 69284 (91.14%) of participants had vaginal delivery (Table 2).

### Maternal/household related characteristics

About 19,693 (25.25%) of participants were from non-educated mothers. More than half, 41085 (54.05%), of participants were employed. About 56727 (74.62%) participants resided in rural areas. Kenya has the highest representation, 17327 (22.79%), and Comoros has the lowest, 2387 (3.41%). More than half of participants, 40948 (53.87%), have normal height; similarly, about 47835 (53.87%) of mothers have normal BMI (Table 3).

### Prevalence of severe stunting among under five in Eastern African Countries

The overall prevalence of severe stunting among children under five years old in 12 Eastern African countries was 8.61% (95% CI: 8.41, 8.80). The highest rates were found in Burundi, at 23.94% (95% CI: 22.87, 25.02), followed by Ethiopia, 16.47% (95% CI: 15.69, 17.24), and the lowest prevalence was found in Kenya, 4.33% (95% CI: 4.03, 4.63). The detailed prevalence for each country is provided using a Forest Tree plot in (Fig 1). Fig 1. Forest Tree plot for the prevalence of severe stunting, in Eastern African countries from DHS 2012–2022.

**Table 2. Child-related/maternity-related characteristics for predicting severe stunting and its determinants among under-five children in Eastern African Countries, 2012–2022 (n = 76019).**

| Variable | Categories | Frequency | Percentage |
|---|---|---|---|
| **Child's sex** | Male | 38176 | 50.22 |
| | Female | 37843 | 49.78 |
| **Child's age** | < 24 | 17804 | 23.42 |
| | 24–36 | 45855 | 60.32 |
| | 37–47 | 12087 | 15.90 |
| | 48–59 | 273 | 0.36 |
| **Place of delivery** | Home | 18962 | 24.94 |
| | Health facility | 57057 | 75.06 |
| **Modes of delivery** | Cesarean | 6735 | 8.86 |
| | Vaginal | 69284 | 91.14 |
| **Types of delivery** | Single | 73945 | 97.27 |
| | Twin | 2074 | 2.73 |
| **PNC visit** | No | 55935 | 73.58 |
| | Yes | 20084 | 26.42 |
| **Exclusive breastfeeding for six months** | No | 43422 | 57.12 |
| | Yes | 32597 | 42.88 |
| **ANC visit** | No | 38297 | 50.38 |
| | Yes | 37722 | 49.62 |
| **Birth order** | 1–3 | 17563 | 23.10 |
| | 4–6 | 37784 | 49.70 |
| | >=7 | 20672 | 27.19 |
| **Child size at birth** | Small | 15561 | 20.47 |
| | Average | 42475 | 55.87 |
| | Large | 17983 | 23.66 |
| **Underweight status of child** | Underweight | 54377 | 71.53 |
| | No underweight | 21642 | 28.31 |

## ML analysis of severe stunting

**Feature selection.** The Boruta algorithm result revealed that all the variables were colored green and presented above the shadomax. Hence, all variables were confirmed to be important by the algorithm and were used for the next analysis, as shown in (S1 Fig 4 in S1 File).

**Balancing the data.** To handle unbalanced data distributions in the training sets, the SMOTE oversampling technique generated 47,407 additional synthetic data for minority classes in the training sets. Hence, the data was changed from unbalanced to balanced distribution for both classes as shown in (Fig 2). Fig 2. Distribution of severe stunting before and after data balancing for the prediction of severe stunting Eastern African countries, 2012–2022.

**Model building/training and model performance evaluation.** Eight ML models were selected and trained to predict severe stunting accurately based on performance metrics, including AUC score, sensitivity, specificity, and accuracy. A stratified 10-fold cross-validation technique was used to compare the performances of predictive models. After comparing the models with unbalanced data, AdaBoost was the best-performing model with an accuracy of 90% and a 0.93 AUC score; however, the unbalanced nature of the outcome variables leads to distorted results. Hence, after balancing the data, RF achieves better performance than the other models with an accuracy of 87% and an AUC score of 0.83. CatBoost was the second-best-performing model with an accuracy of 84% and a 0.82 AUC score, Similarly, RF achieves

**Table 3. Maternal/household related characteristics for predicting severe stunting and its determinants among under-five children in Eastern African Countries, 2012–2022 (n = 76019).**

| Variable | Categories | Frequency | Percentage |
|---|---|---|---|
| **Mother's Education** | No education | 19693 | 25.25 |
| | Primary | 32949 | 43.34 |
| | Secondary & Higher | 6609 | 8.69 |
| **Marital Status** | Single | 23377 | 30.75 |
| | Divorced/separated | 65084 | 85.62 |
| | Married | 4326 | 5.69 |
| **Mother's Occupation** | Not-working | 34934 | 45.95 |
| | Working | 41085 | 54.05 |
| **Father's Education** | No-Education | 17637 | 23.20 |
| | Primary | 32367 | 42.58 |
| | Secondary & Higher | 26015 | 34.22 |
| **Mother's height** | Short | 26561 | 34.94 |
| | Normal | 40948 | 53.87 |
| | Tall | 8510 | 11.19 |
| **Mother's BMI** | Underweight | 13377 | 17.60 |
| | Normal | 47835 | 62.93 |
| | Overweight | 14807 | 19.48 |
| **Age at first birth** | 9–21 | 57660 | 75.85 |
| | 22–34 | 18127 | 23.85 |
| | 35–47 | 232 | 0.31 |
| **Types of Toilets** | Not improved | 39480 | 51.93 |
| | Improved | 36539 | 48 |
| **Wealth status** | Poor | 36163 | 47.57 |
| | Middle | 14097 | 18.54 |
| | Rich | 25759 | 33.8 |
| **Household head sex** | Male | 56990 | 74.97 |
| | Female | 19029 | 25.03 |
| **Family size** | 1-5 | 41933 | 55.16 |
| | 6–12 | 22993 | 30.25 |
| | >12 | 11093 | 14.59 |
| **Fuel type** | Not-Electricity | 71148 | 93.59 |
| | Electricity | 4871 | 6.41 |
| **Insurance Coverage** | No | 69714 | 91.71 |
| | Yes | 6305 | 8.29 |
| **Media exposure** | No | 30305 | 39.87 |
| | Yes | 45714 | 60.13 |
| **Time to get water** | Quick access | 6227 | 8.19 |
| | Took long hour | 67895 | 89.31 |
| | Far | 1897 | 2.50 |
| **Country** | Burundi | 6048 | 7.96 |
| | Ethiopia | 8855 | 11.65 |
| | Kenya | 17327 | 22.79 |
| | Comoros | 2387 | 3.14 |
| | Madagascar | 5778 | 7.60 |
| | Malawi | 5149 | 6.77 |
| | Mozambique | 3733 | 4.91 |
| | Rwanda | 3809 | 5.01 |
| | Tanzania | 4809 | 6.32 |
| | Uganda | 4423 | 5.82 |
| | Zimbabwe | 8746 | 11.51 |
| | Zambia | 4957 | 6.52 |

*(Continued)*

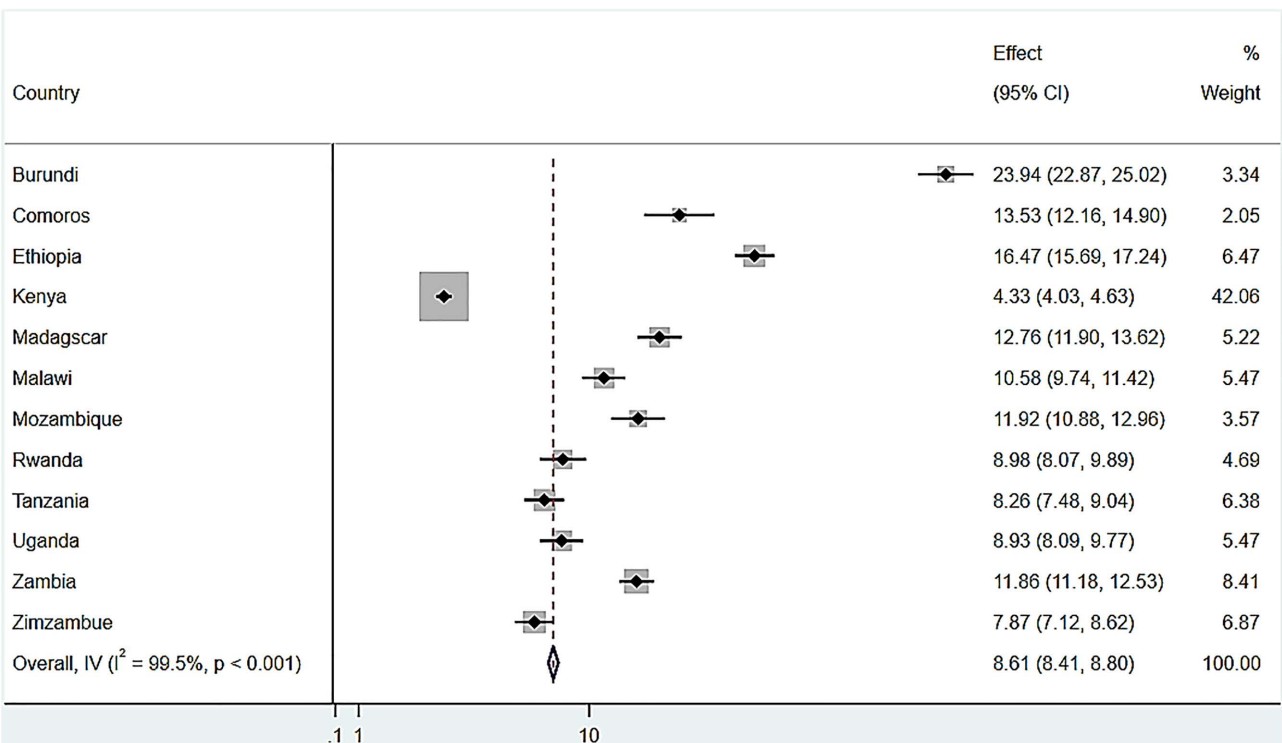

**Table 3.** (Continued)

| Variable | Categories | Frequency | Percentage |
|---|---|---|---|
| **Residence** | Urban | 19292 | 25.38 |
| | Rural | 56727 | 74.62 |
| **Distance to health facility** | A big problem | 48091 | 63.26 |
| | Not a big problem | 27928 | 36.74 |

| Country | | Effect (95% CI) | % Weight |
|---|---|---|---|
| Burundi | | 23.94 (22.87, 25.02) | 3.34 |
| Comoros | | 13.53 (12.16, 14.90) | 2.05 |
| Ethiopia | | 16.47 (15.69, 17.24) | 6.47 |
| Kenya | | 4.33 (4.03, 4.63) | 42.06 |
| Madagscar | | 12.76 (11.90, 13.62) | 5.22 |
| Malawi | | 10.58 (9.74, 11.42) | 5.47 |
| Mozambique | | 11.92 (10.88, 12.96) | 3.57 |
| Rwanda | | 8.98 (8.07, 9.89) | 4.69 |
| Tanzania | | 8.26 (7.48, 9.04) | 6.38 |
| Uganda | | 8.93 (8.09, 9.77) | 5.47 |
| Zambia | | 11.86 (11.18, 12.53) | 8.41 |
| Zimzambue | | 7.87 (7.12, 8.62) | 6.87 |
| Overall, IV ($I^2$ = 99.5%, p < 0.001) | | 8.61 (8.41, 8.80) | 100.00 |

**Fig 1. Forest plot showing the prevalence of severe stunting among under-five children in Eastern African countries using data from DHS 2012–2022.**

the best performance among other models with sensitivity (93%), and specificity (91%), RF also performed best in classifying cases. The performances of each model were presented in (**Table 4**, S1 Fig 5 in S1 File).

### Hyperparameter tuning for the RF classifier

The parameters of RF were optimized with Randomized search techniques, after optimization. The default and optimized values are provided in (**Table 5**).

### Predicting severe stunting using an optimized RF model

After selecting the best model, the RF model was further optimized using Randomized search, the optimized RF model was tested on 15,204 sample test data From 1663 severely stunted cases, the model predicted 1546 cases correctly as severely stunted cases (True Positive), and from 13,541 non-severely stunted cases, the model predicted 12,242 cases

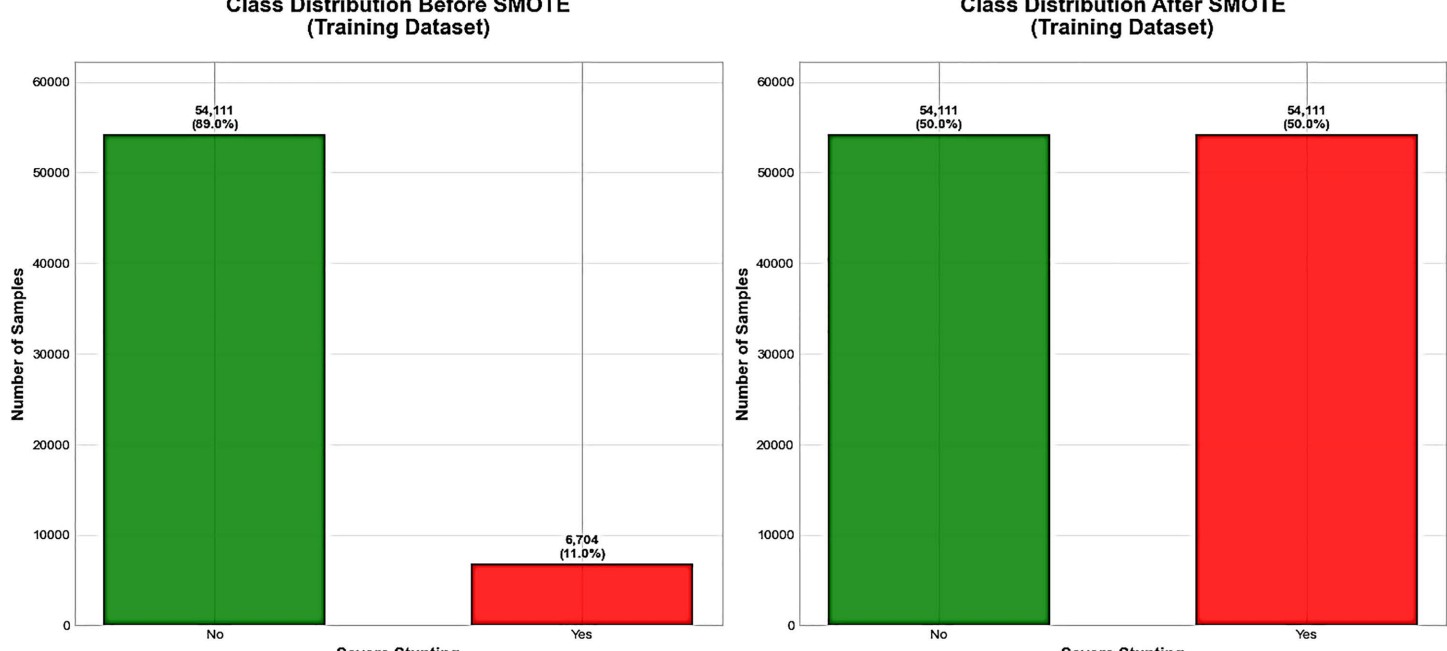

**Fig 2. Distribution of severe stunting before and after data balancing for the prediction of severe stunting Eastern African countries, using data from DHS 2012–2022.**

**Table 4. Model building and performance evaluation using accuracy and AUC scores of balanced and unbalanced training data across different ML models for predictions of severe stunting among children under five, in Eastern African countries from 2012–2022.**

| No | ML Models | Data | Accuracy score (%) | AUC score |
|----|-----------|------|--------------------|-----------|
| 1 | SVM | Unbalanced | 90 | 0.76 |
| | | Balanced | 89 | 0.77 |
| 2 | DT | Unbalanced | 84 | 0.63 |
| | | Balanced | 82 | 0.66 |
| 3 | LR | Unbalanced | 89 | 0.82 |
| | | Balanced | 84 | 0.80 |
| 4 | RF | Unbalanced | 88 | 0.82 |
| | | Balanced | 87 | 0.83* |
| 5 | XGB | Unbalanced | 90 | 0.83 |
| | | Balanced | 84 | 0.81 |
| 6 | KNN | Unbalanced | 89 | 0.79 |
| | | Balanced | 77 | 0.79 |
| 7 | AdaBoost | Unbalanced | 90 | 0.93 |
| | | Balanced | 84 | 0.81 |
| 8 | CatBoost | Unbalanced | 90 | 0.84 |
| | | Balanced | 84 | 0.82 |

**Hint:** * *Maximum performance, AUC – Area under the Curve, DT – Decision Tree, KNN – K-nearest neighbor, LR – Logistic Regression, RF – Random Forest, SVM – Support Vector Machine, XGB – Extreme Gradient Boosting*

**Table 5. Default and optimum hyperparameter values for the RF model for predictions of severe stunting among under five, in Eastern African countries, 2012–2022.**

| No | Hyperparameters | Default value | Optimum value |
|----|-----------------|---------------|---------------|
| 1 | n_estimators | 100 | 375 |
| 2 | max_features | Sqrt of the number of features | 0.77 |
| 3 | min_sample_split | 2 | 7 |
| 4 | min_samples_leaf | 1 | 2 |
| 5 | max_samples | None | 20 |

as non-stunted (True Negative). However, the model misclassified 117 non-severely stunted as severely stunted (False Positive) and 1299 severely stunted cases as not-severely stunted (False Negative), Finally, the model predicted with an accuracy of 87%, precession of 90%, sensitivity (93%), specificity (91%) on the test data as presented in (**Table 6**).

## Model interpretability

**RF-based feature importance.** According to the optimized RF model most significant predictors, 21 variables with higher importance for the model, were further used to identify severe stunting using SHAP. Hence, (variables above minimum importance) were included for further analysis such as underweight status of child, country, child size at birth, family size, birth order, mother's BMI, fathers education, child age, wealth status, mothers education, mother's height, distance to health facility, ANC visit, Toilet types, mothers occupation, marital status, place of delivery, age at firth birth, exclusive breastfeeding, child sex and place of residence are the most important predictors that RF model detects illustrated in (S1 Fig 6 in S1 File).

**SHAP analysis on model prediction.** SHAP global importance score for the prediction of severe stunting using an optimized RF model illustrated that the score was sorted in descending order based on impact on the model output. Variable with more mean absolute SHAP values or variables appear on top of the bar reveal the most important predictors for predicting severe stunting such as: women's who do not practicing exclusive breastfeeding, being from Burundi, children being underweight, lived in poor household,child gender being male, mothers height being short, mothers being underweight, child size at birth being small, women's being delivered in home, mothers education being primary, unimproved toilet, distance to health facility being a big problem were the top most important variables sorted in descending order from higher to lower as illustrated in (Fig 3).

Based on the result presented in the global Beeswarm plot shows that women's who do not practicing exclusive breastfeeding (SHAP value = +0.41), being from Burundi (SHAP value = +0.04), children being underweight (SHAP value = +0.25), lived in poor household (SHAP value = +0.40), child gender being male (SHAP value = +0.23), mothers height being short (SHAP value = +0.03), mothers being underweight (SHAP value = +0.18), child size at birth being small (SHAP value = +0.21), women's being delivered in home(SHAP value = +0.07), mothers education being primary (SHAP value = +0.20), unimproved toilet (SHAP value = +0.06), distance to health facility being a big problem (SHAP

**Table 6. Two-by-two table of optimized RF model for predictions of severe stunting among children under five, in Eastern African countries, 2012–2022.**

| Classes | Predicted class | |
|---------|-----------------|---|
| | Positive prediction | Negative prediction |
| **Positive class** | True positive 1546 | False Negative 1299 |
| **Negative class** | False Positive 117 | True Negative 12,242 |

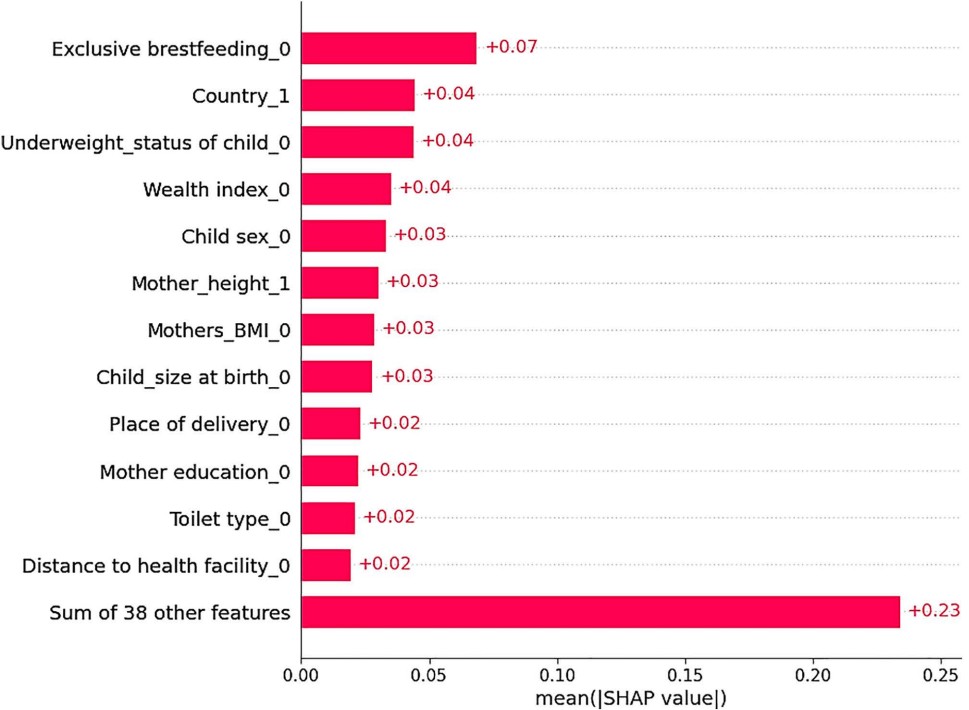

**Fig 3. SHAP global importance plot for the predictions of severe stunting among children under five in Eastern African countries, 2012–2022.** (**Hint**: *exclusive breastfeeding_0 = who do not practicing exclusive breastfeeding, being from country_1 = Burundi, underweight status_0 = children being underweight, wealth index_0 = poor household, child sex_0 = male, mothers height _1, mothers BMI_0 = underweight, child size at birth _0 = small, place of delivery_0 = home delivery, mothers education_0 = primary, toilet type_0 = unimproved, distance to health facility_0 = big problem.*).

value = +0.02), were associated with increase the risk of severe stunting among under five as illustrated in Beeswarm plot (**Fig 4**).

According to local SHAP plots (S1 Fig 7. in S1 File), the red colored bar pointed to the right such as: children being underweight(underweight status = 0), being from Burundi(country = 1), child gender being male(child sex = 0), being poor wealth index(wealth index = 0), unimproved toilet(toilet type = 0), mothers education being primary(mothers education = 1), fathers education being primary(fathers education = 1), mothers height being short(mothers height = 1), child size at birth being small(child size at birth = 0), mothers BMI being underweight (mothers BMI = 0), women's being delivered in home(-place of delivery = 0), distance to health facility being a big problem(distance to health facility = 0), women's being lived in a family size 1–5 (family size = 0), increase the probability of severe stunting for the selected child or they are the risk, whereas the bars in blue color such as: being women's practicing exclusive breastfeeding (exclusive breastfeeding = 1), women's being utilized ANC service(ANC visit = 1),mothers who are employed (Mothers occupation = 1), child who lived in urban(place of residence = 1), child gender being female (child sex = 1) decrease the probability of severe stunting for the selected child or they are protective for severe stunting.

**Association rule mining.** Based on apriori algorithm result seven rules were strongly associated to severe stunting. The determinants are include: country (Burundi), underweight status of child (child is underweight), women's practicing exclusive breastfeeding(women's never experiencing exclusive breastfeeding), ANC visit(women's no receiving ANC services), mothers height (short maternal height), wealth status being poor, mothers education(no education and primary) were the most frequently associated factors to affect severe stunting among under five years children.

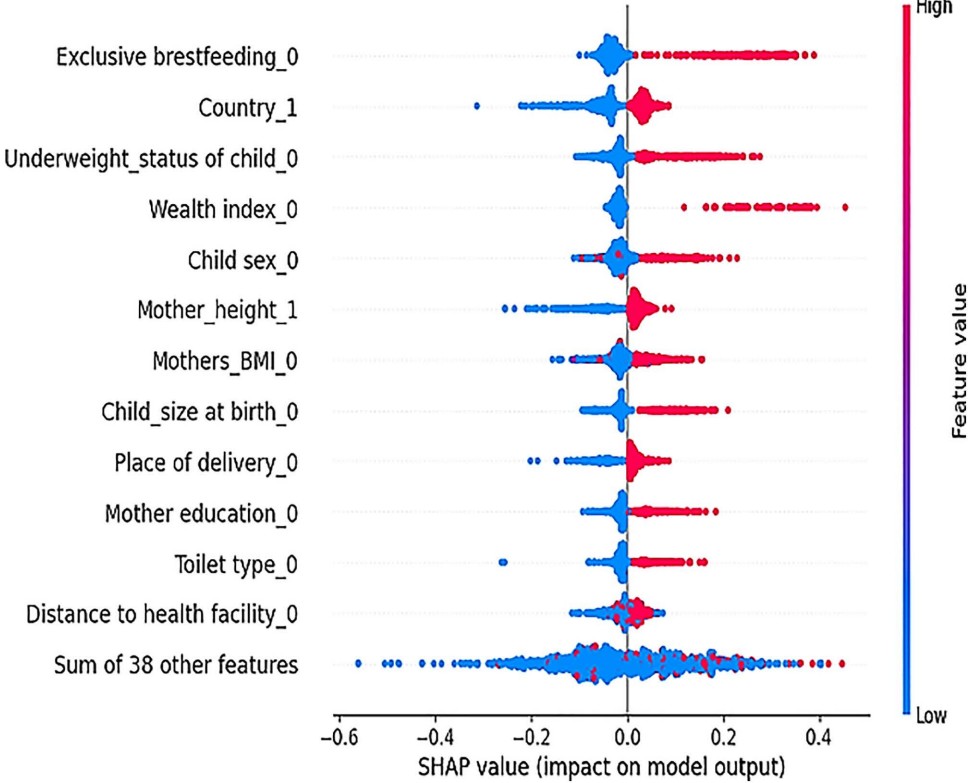

**Fig 4. Beeswarm plot for optimized RF model for the predictions of severe stunting among under five, in Eastern African countries 2012–2022.** (***Hint****: exclusive breastfeeding_0 = who do not practicing exclusive breastfeeding, being from country_1 = Burundi, underweight status_0 = children being underweight, wealth index_0 = poor household, child sex_0 = male, mothers height _1, mothers BMI_0 = underweight, child size at birth _0 = small, place of delivery_0 = home delivery, mothers education_0 = primary, toilet type_0 = unimproved, distance to health facility_0 = big problem.).*

The seven-association rule based on their lift values are listed with their correspondent confidence values and their probabilities of severely stunting are listed below:

Rule 1: (lift 6.39) If country = 1 (Burundi), mother's height = 1 (short), underweight status of child = 0 (child is underweight), and mother's education = 0 (no education), then the probability of severe stunting among children increased by 6.39 times.

Rule 2: (lift 6.51) If country = 1 (Burundi), exclusive breastfeeding = 0 (mothers practicing no breastfeeding), underweight status of child = 0 (child is underweight), and mothers' education = 1 (primary), then the probability of severe stunting among children increased by 6.51 times.

Rule 3: (lift 6.4) If country = 1 (Burundi), mother's height = 1 (short), underweight status of child = 0 (child is underweight), and wealth status = 0 (being poor), then the probability of severe stunting among children increased by 6.4 times.

Rule 4: (lift 6.7) If country = 1 (Burundi), mother's height = 1 (short), underweight status of child = 0 (child is underweight), exclusive breastfeeding = 0 (mothers practicing no breastfeeding), Then the probability of severe stunting among children increased by 6.7 times.

Rule 5: (lift 6.89) If country = 1 (Burundi), wealth status = 0 (being poor), underweight status of child = 0 (child is underweight), exclusive breastfeeding = 0 (mothers practicing no breastfeeding), Then the probability of severe stunting among children increased by 6.89 times.

Rule 6: (lift 6.37) If country = 1(Burundi), ANC visit = 0 (women's not receiving no ANC visit), underweight status of child = 0(child is underweight), exclusive breastfeeding = 0(mothers practicing no breastfeeding), Then the probability of severe stunting among children increased by 6.37 times.

Rule 7: (lift 6.36) If country = 1(Burundi), place of residence = 0 (child lived in rural), underweight status of child = 0 (child is underweight), exclusive breastfeeding = 0(mothers practicing no breastfeeding), Then the probability of severe stunting among children increased by 6.36 times.

## Discussion

The overall prevalence of severe stunting among under-five children in twelve Eastern African countries was found to be 8.61% (95% CI: 8.41, 8.80). Similarly, RF model performed better than all the other models with an accuracy score of 87% and a 0.83 AUC score. Similarly, a study conducted in Zambia [26] identified RF as the best-performing model to predict stunting with an accuracy of 79% [60]. The differences in the accuracy could be due to the size of the dataset used to train the model, as the study from Zambia used 70% of the data for training, and the sample size also significantly varies. Another study conducted in East Africa [46] also identify RF as the best model with an accuracy score of 89%. The difference in accuracy possibly be due to the difference in model optimization techniques, as the previous study did not optimize the model. Similarly, a study conducted in Indonesia revealed that RF were the top utilized model for stunting prediction [49].

Optimized RF model feature importance techniques are utilized to identify variables with higher importance's for the model include: underweight status of child, country, child size at birth, family size, birth order, mother's BMI, fathers education, child age, wealth status, mothers education, mother's height, distance to health facility, ANC visit, Toilet types, mothers occupation, marital status, place of delivery, age at firth birth, exclusive breastfeeding, child sex and place of residence are the most important predictors.

The analysis demonstrated that: Mothers who do not practice exclusive breastfeeding are significantly associated with increasing the risk of severe stunting, which is supported by previous studies reported in SSA [73], Cambodia [74], the reason could be that breast milk has balanced nutrients that are building blocks for the immune system, which can reduce childhood morbidities such as diarrhea and respiratory infections [75]. Furthermore, breast milk provides an optimal infant nutrition formula by promoting antibodies and enzymes needed for development [74].

Likewise, children who lived in Burundi were associated with a higher risk of severe stunting than the other Eastern African nations, the prevalence ranged from the highest rates in Burundi at 23.94% (95% CI: 22.87, 25.02) to the lowest rates found in Kenya, at 4.33% (95% CI: 4.03, 4.63). This finding is supported by another study conducted in SSA, reported that Burundi had the highest prevalence of severe stunting among 25 countries, with a rate of 24.8% (95% CI: 18.8%–25.3%) [13].

In addition, children who are underweight are more likely to experience severe stunting. This result is supported by studies conducted in Gambia [76], and study conducted in Kenya [77] which shows children who are underweight are most likely to be stunted. Similarly a study conducted in 31 SSA gives emphasis to the fact that there is a higher coexistence between underweight, stunting, and wasting among under-fives [78]. This is because there is a positive association between being underweight and stunting, which is often associated with chronic malnutrition, which directly impacts child growth and development [79].

In addition to this, children born from poor families were associated with more severe stunting than children from rich families. This finding was similar to the study conducted in Nepal [17], East Africa [52], Ethiopia [80], which revealed that children in the poor and middle households' wealth level were more likely to be severely stunted. This is due to children who live in poor households typically having poor access to adequate nutrients, safe water, and better hygiene and sanitation. As a result, they are more vulnerable to infections and diseases such as acute respiratory diseases, diarrheal diseases, and intestinal parasites, all of which contribute to severe stunting [81].

Likewise, male children were more likely to be severely stunted than females. This result is supported by previous study findings reported in East Africa [52], SSA [73], LMIC [82]. This could be due to the lower lung maturation among males compared to females that predisposes male children to repeated respiratory infections such as pneumonia, acute respiratory infections, and other airway diseases, which could contribute to the increased risk of stunting among males [83].

As well as, being from short-stature mothers was also associated with severe stunting. This results are supported by previous studies reported in Ethiopia [84],and Kenya [85]. This might be due to the combinational and intergenerational nature of malnutrition through descendants or from families through offspring, intrauterine growth, and development defects [86].

Furthermore, children born from mothers who were underweight are more likely to experience severe stunting than children from mothers with normal body mass index (BMI). There was a similar finding conducted in Ethiopia [18,80,84]. This could be due to maternal undernutrition, which restricts uterine blood flow, impairs growth of the uterus and placenta, and results in poor growth retardation of the uterus, which results in low birth weight, impairs intrauterine growth, and causes growth retardation of the infants [87].

Moreover, children who had a small birth size were associated with severe stunting. This finding is supported by studies conducted in Rwanda [19], Gambia [76], SSA [73], Ethiopia [18], SSA [6], which state that children with low birth weight are more likely to be severely stunted than normal children. This could be due to increased susceptibility of children with low birth weight to infections, mainly diarrheal and lower respiratory infections such as pneumonia, and increased risk of complications including anemia, undernutrition, wasting and chronic lung disorders, fatigue and loss of appetite, and lower immunity to disease compared to children with normal birth weight [52,88].

Additionally, children delivered at home were more likely associated with severe stunting this results are supported by previous studies reported in East Africa, SSA, SSA [13,52,73], and Ethiopia [84], this is because if the children delivered in home there is no additional care provided to the children including post-natal care, basic immunization like iron folate which, is very crucial for the appropriate growth and development of the child as it can prevent several vaccine preventable diseases and supply the child with vital nutrients [89].

Besides, children born to mothers with a lower level of education were more likely to be severely stunted compared to children born to mothers who attained a secondary and higher level of education. It is consistent with the study findings in East Africa [52], Ethiopia [80], SSA [6,73], It could be educated mothers have good knowledge about child health and basic health care services, and an enhanced capacity to recognize childhood illness and seek treatment for their children [90].

Additionally, unimproved toilet facilities were significantly associated with severe stunting, which is supported by studies reported in Ethiopia [38], Indonesia [91], India [92]. Toilets with good sanitation prevent severe stunting by increasing hygienic and environmental friendly way to reducing a number of reservoir for infectious agent, which ultimately decreasing child suitability to infectious pathogens [93].

Lastly, as the distance to healthcare facilities increases, the risk of severe stunting also increases. This finding is supported by a study conducted in Pakistan [33], Ethiopia [94]. This could be because distance to health facility is a barrier that causes the women not to take vital maternal and child care services for addressing the health needs of the children, including PNC visit and ANC visits, basic immunization services [95].

## Strengths and limitations

We used SHAP model interpretability tools, which show how each feature affects the model's predictions. This approach improves transparency compared to traditional black-box ML models like RF. Furthermore, model was tuned to- the best parameters to boost predictive ability. In addition, we used international and standard, scale to measure severe stunting, with the guidelines of WHO to enabled compare, and contrast of findings with different studies. However, the use of SMOTE may limit the findings' applicability to real-world population distributions because the oversampling techniques used to balance the dataset may introduce analytical constraints for the uncommon outcome of severe stunting.

Additionally, the data collected from DHS over a different period interval (2012–2022) also mask important temporal heterogeneities, and the cross-sectional nature of the study design does not show the temporal sequence of the relationship between the variables (cause-and-effect), long term effects or time-specific events such as policy interventions, changes in economics or climate hazards, and public health measures that can affect the prevalence of stunting.

Finally, the secondary nature of the dataset makes it difficult to obtain and analyze more additional necessary variables, due to the absence of detailed dietary diversity and micronutrient intake data in the DHS dataset, which limits our nutritional insights. Furthermore, DHS lacks dietary covariates, preventing analysis of micronutrient deficiencies, including iron, zinc, and vitamin A, which are known drivers of stunting. The absence of nutritional covariates precludes any analysis of micronutrient deficiencies. Therefore, we recommend future research would benefit from longitudinal designs to track causal pathways, incorporation of dietary assessment tools, and validation of ML models on datasets to enhance clinical applicability.

## Conclusion

In our study, demonstrated that RF was the best model to predict severe stunting all the other models, achieved an accuracy and an AUC of 87% and 0.83 respectively. In addition, factors such as women who do not experience breastfeeding, underweight children, children who lived in in Burundi, Children with poor households, Children lived in house with unimproved toilets, mothers being uneducated, child gender being male, mothers short stature, Child born being small, mothers being underweight, home delivery, distance to health facilities being long were significantly associated with increasing the risk of severe stunting among children. To decrease the effects of severe stunting, integrated interventions should provide support for mothers with lower socioeconomic conditions, strengthen maternal education, empower women to practice exclusive breastfeeding, encourage facility deliveries, increase access for households to sanitary facilities, provide education on personal and environmental hygiene, provide mothers with information on the importance of complementary feeding for children as well as for the mothers, and provide near health facilities for mothers and essential care services.

### Implications of the study

The findings of this study carry significant implications for public health policy and nutritional intervention strategies in Eastern Africa. The identification of multiple modifiable risk factors including non-exclusive breastfeeding practices, maternal undernutrition, unimproved sanitation facilities, and limited healthcare access provides a clear roadmap for targeted interventions. Programs should prioritize the first 1,000 days of life, focusing on improving maternal nutrition and ANC care, promoting exclusive breastfeeding practices, and enhancing household sanitation infrastructure. The substantial regional disparities in Burundi underscore the urgent need for context-specific approaches that address unique socioeconomic and healthcare system challenges within the country. Furthermore, the study highlights the critical importance for healthcare systems to integrate nutritional supplements with maternal and child health services, particularly targeting disadvantaged populations through community-based outreach programs. Educational initiatives should focus on improving health literacy, while economic policies should address healthcare accessibility to remote areas.

## Supporting information

**S1 File. Supporting information containing supplementary tables (S1 Tables 1 and 2) and figures (S1 Figs 1–8) that provide additional methodological details, and supporting analyses for the study.**
(DOCX)

## Acknowledgments

The authors are grateful to all the data curation, supervisor, study participants, and Wollo University for their creditable contributions to the success of this study. The authors would also like to thank the Measure of DHS program committee for authorizing the use of the datasets.

## Author contributions

**Conceptualization:** Halid Worku Jemil, Altaseb Beyene Kassaw, Kassahun Dessie Gashu.

**Data curation:** Halid Worku Jemil, Sonia Worku Semayneh.

**Formal analysis:** Halid Worku Jemil.

**Funding acquisition:** Halid Worku Jemil.

**Investigation:** Sonia Worku Semayneh, Kassahun Dessie Gashu.

**Methodology:** Halid Worku Jemil, Altaseb Beyene Kassaw, Kassahun Dessie Gashu.

**Project administration:** Halid Worku Jemil, Sonia Worku Semayneh.

**Software:** Halid Worku Jemil, Altaseb Beyene Kassaw.

**Validation:** Halid Worku Jemil.

**Visualization:** Halid Worku Jemil.

**Writing – original draft:** Halid Worku Jemil, Altaseb Beyene Kassaw.

**Writing – review & editing:** Halid Worku Jemil, Sonia Worku Semayneh, Altaseb Beyene Kassaw, Kassahun Dessie Gashu.

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
