## [Decision Letter · Decision Letter 0]

18 Nov 2025

Dear Dr. Jemil,

Thank you for submitting your manuscript to PLOS ONE. After careful consideration, we feel that it has merit but does not fully meet PLOS ONE’s publication criteria as it currently stands. Therefore, we invite you to submit a revised version of the manuscript that addresses the points raised during the review process.

ACADEMIC EDITOR: Experts in the field have reviewed your manuscript and you are expected to address their comments as early as possible. Thank you. />==============================

We look forward to receiving your revised manuscript.

Kind regards,

Olutosin Ademola Otekunrin

Academic Editor

PLOS ONE

Journal Requirements:

3. PLOS requires an ORCID iD for the corresponding author in Editorial Manager on papers submitted after December 6th, 2016. Please ensure that you have an ORCID iD and that it is validated in Editorial Manager. To do this, go to ‘Update my Information’ (in the upper left-hand corner of the main menu), and click on the Fetch/Validate link next to the ORCID field. This will take you to the ORCID site and allow you to create a new iD or authenticate a pre-existing iD in Editorial Manager

4. Please amend the manuscript submission data (via Edit Submission) to include author Sonia Worku Semayneh, Altaseb Beyene Kassaw, Kassahun Dessie Gashu.

Reviewers' comments:

Reviewer's Responses to Questions

1. Is the manuscript technically sound, and do the data support the conclusions?

Reviewer #1: No

Reviewer #2: Yes

2. Has the statistical analysis been performed appropriately and rigorously?

Reviewer #1: No

Reviewer #2: Yes

3. Have the authors made all data underlying the findings in their manuscript fully available?

Reviewer #1: Yes

Reviewer #2: Yes

4. Is the manuscript presented in an intelligible fashion and written in standard English?

Reviewer #1: No

Reviewer #2: Yes

Reviewer #1: Reviewer report for PONE-D-25-40393-1 (Predicting Severe Stunting and its Determinants among Under-five Children in Eastern African Countries using the Demographic and Health Survey: A Machine Learning Algorithm)

Summary recommendation

•Recommend: Major Revision

•Rationale: The study addresses an important RMNCH topic using a large, multinational DHS dataset and applies multiple machine learning models with SHAP interpretability. However, critical issues remain regarding methodological choices (data balancing with SMOTE, incorporation of DHS complex survey design, and external validity across 12 countries), reproducibility, and interpretation of results. These issues must be resolved with robust sensitivity analyses, transparent reporting, and careful framing of conclusions as associations rather than causal inferences.

A. Major issues

1.Data provenance, harmonization, and study design

•Comment: The manuscript uses DHS data from 12 Eastern African countries (2012–2022). While leveraging large, comparable datasets is valuable, cross-country DHS data vary in survey year windows, sampling frames, and measurement harmonization. The manuscript should clearly document how each variable was harmonized across countries, and provide a country-by-country appendix or table that shows the exact survey year, sample size, and variable coding for key predictors.

•Recommendation: Include a comprehensive data dictionary mapping each predictor to its DHS source, harmonization decisions, and rationale. Add a flow diagram showing inclusion/exclusion criteria and the final analytical sample by country and year.

2.Handling of complex survey design and weights

•Comment: DHS data are collected with complex survey designs (weights, primary sampling units, strata). The modeling workflow should account for these design features to obtain unbiased estimates and valid generalizability. Whether survey weights and clustering were incorporated into model training, evaluation, and calibration, or whether a design-based sensitivity analysis was performed, is unclear.

•Recommendation: Provide a detailed description of how weights, clustering, and stratification were integrated into the machine learning pipeline. If weights were not used, justify why and discuss potential biases. Consider presenting results with and without weights, performing a design-aware modeling approach, and reporting any differences.

3.Data balancing with SMOTE and model validity

•Comment: The study balances the data with SMOTE to address class imbalance before model training. While SMOTE can improve predictive performance on imbalanced data, it may introduce artifacts that affect external validity and lead to optimistic performance estimates if not carefully validated.

•Recommendation: Present sensitivity analyses comparing SMOTE-balanced models to alternative approaches (e.g., class weighting, threshold tuning, stratified sampling) and to models trained on the original imbalanced data. Report performance metrics (AUC, sensitivity, specificity, accuracy) on held-out data without synthetic samples to gauge generalizability.

4.Model evaluation and overfitting

•Comment: Eight models were trained with 10-fold cross-validation, but it is unclear whether a separate hold-out test set was used beyond cross-validation. Without an independent test set, there is a risk of optimistic bias in reported performance.

•Recommendation: If feasible, include a truly independent test set (e.g., hold-out countries or years not used in training) to provide an unbiased assessment of predictive performance. If not possible, provide rigorous cross-validation design details (fold stratification by country/year, nesting of hyperparameter tuning within folds) and discuss potential overfitting.

5.Interpretability and SHAP results

•Comment: SHAP explains model predictions and is a strength of the study. However, interpretation should be cautious about across-country heterogeneity and potential confounding factors not captured by the predictors.

•Recommendation: Present SHAP results separately by country where feasible or report whether global SHAP patterns are consistent across countries. Include a supplementary table/figure showing country-specific top predictors. Discuss limitations in attributing causal interpretation to SHAP-derived associations.

6.Causal language and policy implications

•Comment: The manuscript occasionally implies determinants cause severe stunting. Given the cross-sectional design, causal claims should be avoided.

•Recommendation: Reframe conclusions to emphasize associations and potential mechanisms, with explicit caveats about temporality and confounding. When discussing policy implications, frame them as hypotheses for targeted interventions that require experimental or longitudinal validation.

7.Reproducibility and code availability

•Comment: Reproducibility is essential, particularly for ML workflows on DHS data. The manuscript should provide access to code and a detailed computational appendix.

•Recommendation: Include a dedicated reproducibility section with links to a code repository (DOI or URL), a README describing dependencies, data processing steps, model training, and SHAP analysis, and instructions to reproduce the results. Clearly state any data restrictions and how to access the DHS data used (with proper permissions).

8.Reporting clarity and structure

•Comment: The abstract and some sections should more precisely reflect the study design and key results. The abstract currently reports multiple numerical performance metrics without context for generalizability.

•Recommendation: Update the abstract to include study design (cross-sectional DHS data across 12 countries, 2012–2022), primary outcome (severe stunting), main modeling approach (eight ML models with SHAP), and the most robust finding(s). Ensure tables and figures have self-contained captions and align with the text.

B. Minor issues

1.Terminology and variable definitions

•Comment: Ensure consistent terminology for predictors (e.g., “birth size,” “birth weight,” “birth order”) and provide explicit category definitions.

2.Figures and tables

•Comment: Some figures (e.g., SHAP plots, ROC curves, feature importance) should have high-resolution formats and clear axis labels.

•Recommendation: Ensure figure captions describe what is being shown, axis scales, and units. Consider including a supplementary figure that summarizes country-level results.

3.Data availability statement

•Comment: Data from DHS is controlled; provide a data availability statement that describes access permissions and any restrictions, plus details on code availability.

4.Ethical considerations

•Comment: The manuscript uses publicly available, de-identified DHS data. Include a brief ethics statement confirming that the study used secondary data and did not involve direct human subject contact, with reference to DHS approvals.

Reviewer #2: I have gone the manuscript thoroughly, and my review feedback as followed.

1.The Title of the manuscript could be more innovative and more concise.

2.The best model Randomforest estimated an accuracy was 92%, which is very high accuracy and sensitivity as on a few features or variables. Author should recheck this model performance. You should use only test dataset to test the model performance instead of all datasets (training and test). Accuracy estimate with both data will be more informative and reliable.

3.Burundi analysis is good for cumulative features scores. I recommend authors to analyze feature important score and present a bar graph with scores for each feature.

4.Machine learning models predict a score for positive and negative classes and then author classify the scores based on a cutoff value (like 0.5) which is depend on the best combination of a set of factors. However, the best set of factors not ensure causal factors. Authors may find out more vulnerable children by using independence conditional estimator (ICE) analysis.

5.In SHAP analysis, I used country as a predictor and it highlights most important factors. It may a cluster factor but considered as a key important factor is not possible. Authors have used some categorical variable as a continuous variable and figure out the most important factors. It difficult to interpret. Authors should use as categorical variables in the models like Wealth index, mother education, toilet types etc.

Do you want your identity to be public for this peer review? For information about this choice, including consent withdrawal, please see our Privacy Policy

Reviewer #1: No

Reviewer #2: Yes: Probir Kumar Ghosh

---

## [Author Response · Author response to Decision Letter 1]

1 Dec 2025

Date: 27 November 2025

To: PLOS ONE, Editor-in-Chief

Subject: a point-by-point response to reviewer’s comments

Manuscript title: Predicting Severe Stunting and its Determinants among Under-five in Eastern African Countries: A Machine Learning Algorithms. PONE-D-25-40393-1

EMID: e8756fa6ee451d93

Dear Dr. Olutosin Ademola Otekunrin,

We are very grateful for the comments of the reviewers on our manuscript, which we believe have improved our manuscript for publication. Following the valuable comments and recommendations of the reviewers, we have revised the manuscript and hereby submit the revised manuscript along with a point-by-point response to the reviewers for your consideration.

Dear editor,

We would like to sincerely thank you for facilitating the review process and for your efforts in securing such knowledgeable and professional reviewers for our manuscript. Your support has been invaluable in guiding us through the revision process, and we greatly appreciate your assistance in ensuring the quality of the review.

Kindest regards,

Sincerely yours,

A point-by-point response to reviewer 1 comments

Reviewer 1

Dear reviewer, thank you for your constructive and valuable comments and concerns. We have accepted your comments. Following your valuable comments and recommendations, we have extensively revised the manuscript.

“Summary recommendation

•Recommend: Major Revision

•Rationale: The study addresses an important RMNCH topic using a large, multinational DHS dataset and applies multiple machine learning models with SHAP interpretability. However, critical issues remain regarding methodological choices (data balancing with SMOTE, incorporation of DHS complex survey design, and external validity across 12 countries), reproducibility, and interpretation of results. These issues must be resolved with robust sensitivity analyses, transparent reporting, and careful framing of conclusions as associations rather than causal inferences.

A. Major issues

1.Data provenance, harmonization, and study design

•Comment: The manuscript uses DHS data from 12 Eastern African countries (2012–2022). While leveraging large, comparable datasets is valuable, cross-country DHS data vary in survey year windows, sampling frames, and measurement harmonization. The manuscript should clearly document how each variable was harmonized across countries, and provide a country-by-country appendix or table that shows the exact survey year, sample size, and variable coding for key predictors.

•Recommendation: Include a comprehensive data dictionary mapping each predictor to its DHS source, harmonization decisions, and rationale. Add a flow diagram showing inclusion/exclusion criteria and the final analytical sample by country and year“.

Response

Dear reviewer we have appreciated your comments and we are thank full, to ensure model generalizability and mitigate overfitting, the dataset was spitted into training (80%) and a hold-out test set (20%) before doing analysis. Model evaluation and hyperparameter tuning were performed using 10-fold stratified K-fold cross-validation technique on the training set with regularization parameters. Then the final model, selected based on cross-validation performance evaluation on testing set (20%).

Revised location: line number 140 - 147.

To handle unbalanced data distributions in the training sets, the SMOTE oversampling technique generated 47,407 additional synthetic data for minority classes in the training sets. Hence, the data was changed from unbalanced to balanced distribution for both classes as shown in (Figure 5).

And the flow diagram also provided in the supplementary file as well

2. “Handling of complex survey design and weights

•Comment: DHS data are collected with complex survey designs (weights, primary sampling units, strata). The modeling workflow should account for these design features to obtain unbiased estimates and valid generalizability. Whether survey weights and clustering were incorporated into model training, evaluation, and calibration, or whether a design-based sensitivity analysis was performed, is unclear.

•Recommendation: Provide a detailed description of how weights, clustering, and stratification were integrated into the machine learning pipeline. If weights were not used, justify why and discuss potential biases. Consider presenting results with and without weights, performing a design-aware modeling approach, and reporting any differences.”

Response

We are thank full for and appreciated your comments and we applied weight adjustment before analysis based on your valuable comments we accordingly revised the documents in order to ensure the complexity arise from the sampling design weighted scaling were applied befor douing analysis to insure representativeness..

Revised location: line number 99 – 102.

Before doing the analysis, weight adjustments were applied to handle the complexity of sampling design and to ensure representativeness using Stata version 17 for. We adjusted the data for both the outcome and predictors using DHS sample weights (v005/1,000,000) in all descriptive analyses. This helps to correct for unequal probability of selection and to ensure a national representativeness of samples. Additionally, the data were preprocessed, missing values were managed before conducting analysis.

3. Data balancing with SMOTE and model validity

•Comment: The study balances the data with SMOTE to address class imbalance before model training. While SMOTE can improve predictive performance on imbalanced data, it may introduce artifacts that affect external validity and lead to optimistic performance estimates if not carefully validated.

•Recommendation: Present sensitivity analyses comparing SMOTE-balanced models to alternative approaches (e.g., class weighting, threshold tuning, and stratified sampling) and to models trained on the original imbalanced data. Report performance metrics (AUC, sensitivity, specificity, accuracy) on held-out data without synthetic samples to gauge generalizability.

Response

Thank you for your valuable comment, based on your comment we accordingly revised it by incorporating K-fold stratified sampling techniques along with SMOTE In order to train the prediction function, k-fold splits all observations into equal-sized sample groups called folds and k-1 folds. The remaining fold is then used for testing k times in a row. The average of the values calculated in the loop is the k-fold cross-validation performance metric. This study employed a 10-fold CV technique to train and test data since 10-fold CV prevents data waste, and enables model to train on most characteristics and a smaller amount of data for testing. It is an efficient way to enhance model performance, the before/after class balancing were also reported the performance metrics AUC, sensitivity, specificity, accuracy in the supplementary Figure 5 and Table 1.

Revised location: line number 133 – 138.

4. Model evaluation and overfitting

•Comment: Eight models were trained with 10-fold cross-validation, but it is unclear whether a separate hold-out test set was used beyond cross-validation. Without an independent test set, there is a risk of optimistic bias in reported performance.

•Recommendation: If feasible, include a truly independent test set (e.g., hold-out countries or years not used in training) to provide an unbiased assessment of predictive performance. If not possible, provide rigorous cross-validation design details (fold stratification by country/year, nesting of hyperparameter tuning within folds) and discuss potential overfitting.

Response

To make good predictive models, and to mitigate the effect of model overfitting. We employed a 10-fold CV technique to train and test the data since 10-fold CV prevents data waste and provides a lot of data for training, and we gain reliable results since most of the data (9 times) was used for training. The model learned from the training data split of 20/80. Data was split into training (80%) and testing sets (20%) to evaluate the model performances

Revised location: line number 202 – 217.

Data was split into training and testing sets by allocating 80% (54,111) of the data for training and 20% (15,204) of the data for testing to evaluate the model. This study employed a 10-fold CV technique to train and test data since 10-fold CV prevents data waste, and enables model to train on most characteristics and a smaller amount of data for testing. It is an efficient way to enhance model performance

5. Interpretability and SHAP results

•Comment: SHAP explains model predictions and is a strength of the study. However, interpretation should be cautious about across-country heterogeneity and potential confounding factors not captured by the predictors.

•Recommendation: Present SHAP results separately by country where feasible or report whether global SHAP patterns are consistent across countries. Include a supplementary table/figure showing country-specific top predictors. Discuss limitations in attributing causal interpretation to SHAP-derived associations.

Response

Dear reviewer, we are grateful for your comment, based on your comment we make clearer understanding for readers by providing it as a limitation section and recommend future researches to include the remain variables for consideration.

Revised location: line number 569 – 569.

Additionally, the data collected from DHS over a different period interval (2012–2022) also mask important temporal heterogeneities, and the cross-sectional nature of the study design does not show the temporal sequence of the relationship between the variables (cause-and-effect), long term effects or time-specific events such as policy interventions, changes in economics or climate hazards, and public health measures that can affect the prevalence of stunting.

6. “Causal language and policy implications

•Comment: The manuscript occasionally implies determinants cause severe stunting. Given the cross-sectional design, causal claims should be avoided.

•Recommendation: Reframe conclusions to emphasize associations and potential mechanisms, with explicit caveats about temporality and confounding. When discussing policy implications, frame them as hypotheses for targeted interventions that require experimental or longitudinal validation.”

Response

Dear reviewer thank you for raising such issues following your valuable comment we provide your comments in a concise and brief manner by harmonizing and clearly reporting the associational findings in conclusions and as well as to increase interpretation in policy implication of the study. And clearly discussing in the limitation section for future studies to include longitudinal desgn.

Revised location: line number 565 – 569.

Additionally, the data collected from DHS over a different period interval (2012–2022) also mask important temporal heterogeneities, and the cross-sectional nature of the study design does not show the temporal sequence of the relationship between the variables (cause-and-effect), long term effects or time-specific events such as policy interventions, changes in economics or climate hazards, and public health measures that can affect the prevalence of stunting.

Revised location: line number 578 – 675.

Conclusion In our study, demonstrated that RF was the best model to predict severe stunting all the other models, achieved an accuracy and an AUC of 87% and 0.83 respectively. In addition, factors such as women who do not experience breastfeeding, underweight children, children who lived in in Burundi, Children with poor households, Children lived in house with unimproved toilets, mothers being uneducated, child gender being male, mothers short stature, Child born being small, mothers being underweight, home delivery, distance to health facilities being long were significantly associated with increasing the risk of severe stunting among children. To decrease the effects of severe stunting, integrated interventions should provide support for mothers with lower socioeconomic conditions, strengthen maternal education, empower women to practice exclusive breastfeeding, encourage facility deliveries, increase access for households to sanitary facilities, provide education on personal and environmental hygiene, provide mothers with information on the importance of complementary feeding for children as well as for the mothers, and provide near health facilities for mothers and essential care services.

Implications of the Study The findings of this study carry significant implications for public health policy and nutritional intervention strategies in Eastern Africa. The identification of multiple modifiable risk factors including non-exclusive breastfeeding practices, maternal undernutrition, unimproved sanitation facilities, and limited healthcare access provides a clear roadmap for targeted interventions. Programs should prioritize the first 1,000 days of life, focusing on improving maternal nutrition and ANC care, promoting exclusive breastfeeding practices, and enhancing household sanitation infrastructure. The substantial regional disparities in Burundi underscore the urgent need for context-specific approaches that address unique socioeconomic and healthcare system challenges within the country. Furthermore, the study highlights the critical importance for healthcare systems to integrate nutritional supplements with maternal and child health services, particularly targeting disadvantaged populations through community-based outreach programs. Educational initiatives should focus on improving health literacy, while economic policies should address healthcare accessibility to remote areas.

7. “Reproducibility and code availability

•Comment: Reproducibility is essential, particularly for ML workflows on DHS data. The manuscript should provide access to code and a detailed computational appendix.

•Recommendation: Include a dedicated reproducibility section with links to a code repository (DOI or URL), a README describing dependencies, data processing steps, model training, and SHAP analysis, and instructions to reproduce the results. Clearly state any data restrictions and how to access the DHS data used (with proper permissions).”

Response

Thank you for your valuable comments and recommendation, we accordingly revised the documents in order to ensure the DHS protocols and privacy as well as the github link to increase code reproducibility.

Revised location: line number 631 – 635.

Availability of data and materials

The data was presented in the study are publicly available at the DHS program website https://www.DHSprogram.com and additional files are provided in the supplementary material; further inquiries can be directed to the corresponding author. In addition, code and preprocessing scripts are available at Upload files · Halwor/code.

8. Reporting clarity and structure

•Comment: The abstract and some sections should more precisely reflect the study design and key results. The abstract currently reports multiple numerical performance metrics without context for generalizability.

•Recommendation: Update the abstract to include study design (cross-sectional DHS data across 12 countries, 2012–2022), primary outcome (severe stunting), main modeling approach (eight ML models with SHAP), and the most robust finding(s). Ensure tables and figures have self-contained captions and align with the text.

Response

Dear reviewer, we appreciate your comment and invaluable suggestion. We have revised and addressed your notable comments by including study design, outcome, main modeling approach and making the methods in the abstract more clearly by moving tables and figures to the supplementary file as well as by merging and shortening methods sections that are too long.

Revised location: line number 31 – 53.

Methods: cross-sectional study was conducted using DHS data from 2012–2022 in East Africa. 136,074 children were the source populations, and 76,019 children were the study population. Data were analyzed using Python version 3.7 and R version 4.3.3 for data preprocessing, modeling, and statistical analysis. Model performance was evaluated using accuracy and AUC. Furthermore, the SHAP anal

---

## [Decision Letter · Decision Letter 1]

17 Dec 2025

Predicting Severe Stunting and its Determinants among Under-five in Eastern African Countries: A Machine Learning Algorithm.

PONE-D-25-40393R1

Dear Dr. Jemil,

We’re pleased to inform you that your manuscript has been judged scientifically suitable for publication and will be formally accepted for publication once it meets all outstanding technical requirements.

Kind regards,

Olutosin Ademola Otekunrin

Academic Editor

PLOS One

Additional Editor Comments (optional):

Reviewers' comments:

Reviewer's Responses to Questions

**Comments to the Author**

Reviewer #2: All comments have been addressed

2. Is the manuscript technically sound, and do the data support the conclusions?

Reviewer #2: Yes

3. Has the statistical analysis been performed appropriately and rigorously?

Reviewer #2: Yes

4. Have the authors made all data underlying the findings in their manuscript fully available?

Reviewer #2: Yes

5. Is the manuscript presented in an intelligible fashion and written in standard English?

Reviewer #2: Yes

Reviewer #2: Dear Authors,

Thanks you, for addressing all comments correctly. I recommend you that you should study on causal analysis in the stunting because machine learning predictors do not ensure causal relationship between factors and outcome.

Thanks,

Probir

**Do you want your identity to be public for this peer review?** For information about this choice, including consent withdrawal, please see our Privacy Policy

Reviewer #2: **Yes: ** Probir Kumar Ghosh

---

## [Editor Report · Acceptance letter]

PONE-D-25-40393R1

PLOS One

Dear Dr. Jemil,

I'm pleased to inform you that your manuscript has been deemed suitable for publication in PLOS One. Congratulations! Your manuscript is now being handed over to our production team.

Kind regards,

on behalf of

Dr. Olutosin Ademola Otekunrin

Academic Editor

PLOS One